# Comparison of Creatinine-, Cystatin C-, and Combined Creatinine–Cystatin C-Based Equations for Estimating Glomerular Filtration Rate: A Real-World Analysis in Patients with Chronic Kidney Disease

**DOI:** 10.3390/ijms27010364

**Published:** 2025-12-29

**Authors:** Joško Osredkar, Iza Klemenčič, Kristina Kumer, Jernej Pajek, Bojan Knap

**Affiliations:** 1Institute of Clinical Chemistry and Biochemistry, University Medical Centre Ljubljana, Zaloška cesta 2, 1000 Ljubljana, Slovenia; josko.osredkar@kclj.si (J.O.);; 2Faculty of Pharmacy, University of Ljubljana, Aškerčeva 7, 1000 Ljubljana, Slovenia; iza.klemencic@gmail.com; 3Department of Nephrology, Division of Internal Medicine, University Medical Centre Ljubljana, Zaloška cesta 2, 1000 Ljubljana, Slovenia; jernej.pajek@kclj.si; 4Faculty of Medicine, University of Ljubljana, Vrazov trg 2, 1000 Ljubljana, Slovenia

**Keywords:** chronic kidney disease, glomerular filtration rate, eGFR, CKD-EPI, creatinine, cystatin C, GFR classification, diagnostic accuracy

## Abstract

The estimated glomerular filtration rate (eGFR) is a cornerstone of kidney function assessment. Widely used Chronic Kidney Disease Epidemiology Collaboration (CKD-EPI) equations based on serum creatinine (eGFR_cr_), cystatin C (eGFR_cysC_), or both (eGFR_cr-cysC_) are influenced by non-glomerular filtration rate (GFR) factors, and their performance may vary across clinical contexts. We retrospectively analyzed 435 adult patients with simultaneous serum creatinine and cystatin C measurements. eGFR was calculated using CKD-EPI 2021 (creatinine), CKD-EPI 2012 (cystatin C), and CKD-EPI 2021 (combined) equations. Patients were classified into Kidney Disease: Improving Global Outcomes (KDIGO) GFR categories (G1–G5), and discrepancies between equations were identified. 44 patients (10.1%) showed discordant GFR categorization across all three equations and underwent detailed clinical assessment. 16 of the 44 discordant cases had clinically confirmed chronic kidney disease (CKD). The combined equation aligned with the clinical diagnosis in all CKD cases. eGFR_cr_ overestimated kidney function in 10/16 patients, while eGFR_cysC_ produced lower values in 8/16, consistent with early CKD but potentially influenced by inflammation or obesity. Reclassification occurred in 9/16 patients when switching from eGFR_cr_ to eGFR_cr-cysC_, including four who shifted from G2 to G3a–G4. A significant difference was observed between eGFR_cr_ and eGFR_cr-cysC_ (*p* < 0.05). The combined CKD-EPI equation demonstrated the best clinical concordance, supporting its broader use when diagnostic accuracy is essential.

## 1. Introduction

Chronic kidney disease (CKD) is a growing global health concern, affecting over 850 million individuals worldwide and associated with significantly increased cardiovascular morbidity, mortality, and health care costs [1,2,3,4]. Early identification and accurate staging of CKD are essential for optimizing management, adjusting nephrotoxic or renally-excreted medications, and improving patient outcomes.

The glomerular filtration rate (GFR) is widely accepted as the best overall indicator of kidney function, yet its direct measurement (mGFR) using exogenous filtration markers (e.g., inulin, iohexol) or measured creatinine clearance (requiring urine creatinine, serum creatinine, and urinary volume) is labor-intensive, expensive, and rarely performed in routine practice [5,6,7]. Instead, estimated GFR (eGFR) equations based on endogenous serum markers, primarily serum creatinine (S_cr_), cystatin C (S_cysC_), or both, are commonly used in clinical and laboratory settings [8,9].

Among the most widely adopted formulas are the Chronic Kidney Disease Epidemiology Collaboration (CKD-EPI) equations, which include creatinine-only (CKD-EPI 2021), cystatin C-only (CKD-EPI 2012), and the combined creatinine–cystatin C equation (CKD-EPI 2021). Each has advantages and limitations, particularly related to non-GFR determinants of the serum markers. S_cr_ is influenced by age, sex, muscle mass, diet, and tubular secretion, while cystatin C levels can be affected by inflammation, corticosteroid use, smoking, and metabolic conditions [10,11,12,13,14,15,16].

Several studies have suggested that cystatin C may allow earlier detection of kidney dysfunction, especially in cases where S_cr_ remains within the normal range despite underlying impairment [17,18]. Recent literature and studies advocate incorporating multiple serum or plasma markers of GFR into eGFR to improve accuracy and minimize the influence of non-GF factors on individual markers [3,19]. Moreover, combined creatinine–cystatin C equations have been shown to improve accuracy and prognostic value, especially in patients with comorbidities or atypical body composition [3,19,20,21].

Nevertheless, guidance on when to preferentially use these equations in routine practice remains limited. In particular, real-world comparative data linking eGFR estimates to clinical diagnoses are scarce.

## 2. Results

### 2.1. Agreement and Discrepancy Among eGFR Equations

Of the 435 included patients, 44 patients (10.1%) were assigned to different Kidney Disease: Improving Global Outcomes (KDIGO) GFR categories depending on which equation was used (Figure 1 and Table 1). The greatest variation was observed between CKD-EPI 2021 and CKD-EPI 2012, where cystatin C-based equations tended to classify patients into more advanced CKD stages, particularly in G1 and G5 categories. In contrast, the combined equation generally yielded intermediate values, often mitigating the discrepancy seen between the other two methods.

### 2.2. Subgroup Analysis of Discrepant Cases

From the 44 patients with GFR category discrepancies, 16 patients with confirmed CKD were identified and analyzed in detail (Table 2). The remaining 28 cases included patients with acute kidney injury (AKI), urinary tract infections (UTIs), or insufficient clinical documentation. We systematically screened medication histories for agents known to interfere with biomarker handling. Among the discordant cases, we specifically checked for the use of trimethoprim (which inhibits tubular creatinine secretion) and high-dose corticosteroids (which increase cystatin C production).


**Key observations:**
In all 16 CKD cases, the combined equation most closely reflected the clinical diagnosis, based on nephrology notes, comorbidities, and the presence of albuminuria.The creatinine-based equation frequently overestimated renal function, especially in older adults or patients with reduced muscle mass.The cystatin C-based equation often showed lower GFR values, particularly in patients with inflammatory states or obesity, aligning with early CKD detection but potentially affected by non-GFR factors.


### 2.3. Acute and Infection-Related Cases

Elevated S_cr_, likely due to tubular secretion inhibition, leading to underestimation of GFR by the creatinine-based equation.Elevated S_cysC_, possibly reflecting inflammation.

Among three patients with acute urinary tract infections, trimethoprim use was documented in all cases. These patients had:The combined equation provided a more balanced estimate, consistent with the overall clinical condition.

In five AKI cases, cystatin C increased earlier and more significantly than creatinine, indicating its superior sensitivity for detecting rapid renal function decline.

### 2.4. Statistical Comparison of Equations

As shown in Figure 2, among the 16 patients with verified CKD:Mean eGFR values were significantly different across equations (*p* < 0.05).Two- tailed paired *t*-test confirmed statistically significant differences between eGFR_cr_ and eGFR_cr-cysC_ (*p* = 0.006).Wilcoxon tests did not confirm statistically significant differences between eGFR_cysC_ and eGFR_cr-cysC_ (*p* = 0.0553) and between eGFR_cr_ and eGFR_cr-cysC_ (*p* = 0.0285).

### 2.5. Clinical Relevance

In 9 of the 16 CKD patients (56%), using eGFR_cr-cysC_ led to a different G-category than eGFR_cr_, with potential implications for:Therapeutic decision-making (e.g., drug dosing);Nephrology referral timing;Risk stratification.

In 4 patients, the reclassification was from G2 to G3a–G4, a clinically significant shift for CKD diagnosis.

## 3. Discussion

In this retrospective study, we compared three commonly used equations for eGFR–CKD-EPI 2021 based on creatinine (eGFR_cr_), CKD-EPI 2012 based on cystatin C (eGFR_cysC_), and CKD-EPI 2021 based on a combination of creatinine and cystatin C (eGFR_cr-cysC_)–in a cohort of 435 adult patients. Our findings demonstrate that the choice of equation significantly affects GFR categorization, with implications for the diagnosis and staging of CKD.

### 3.1. Clinical Consequences of Equation Selection

One of the most important findings was that 10% of patients were classified into different KDIGO GFR categories depending on the equation used. This degree of reclassification is clinically meaningful, as KDIGO staging guides not only diagnostic labeling but also timing of nephrology referral, therapeutic interventions, and monitoring frequency [5,6]. We observed that eGFR_cr_ frequently overestimated renal function, especially in elderly patients or those with reduced muscle mass—an expected consequence of creatinine’s dependence on muscle metabolism [10,22]. Conversely, eGFR_cysC_ tended to yield lower eGFR values, which aligned more closely with clinical diagnoses in early CKD, but may also reflect non-GFR influences, such as systemic inflammation [23]. The combined equation (eGFR_cr-cysC_) consistently provided eGFR estimates that were most concordant with the clinical diagnosis of CKD, reducing the number of misclassified cases and aligning better with nephrology assessments. These findings support previous research suggesting that the combined equation improves eGFR accuracy and risk prediction in diverse patient populations [7,20,24].

### 3.2. Importance of Patient-Specific Context

Our results underscore the need for patient-centered selection of eGFR equations. In patients with atypical body composition, dietary habits, or comorbidities (e.g., corticosteroid use, infection), relying solely on eGFR_cr_ may lead to misleading interpretations. For instance, in cases of urinary tract infections treated with trimethoprim, creatinine levels were acutely elevated due to tubular secretion inhibition, falsely suggesting impaired renal function. In such settings, the use of eGFR_cr-cysC_ yielded a more reliable estimate. In contrast, in AKI cases, cystatin C increased earlier than creatinine, consistent with studies indicating that S_cysC_ may be more responsive to acute changes in GFR [17,25]. While cystatin C alone may thus be advantageous in dynamic or unstable renal states, the combined equation still offered a balanced interpretation.

It is important to distinguish between physiological GFR changes and biomarker interference. In patients with inflammation (elevated CRP) or obesity, cystatin C levels may rise independent of filtration changes. In our study, this likely resulted in eGFR_cysC_ underestimating the true renal function in specific cases. Conversely, eGFR_cr_ likely overestimated function in sarcopenic patients due to low creatinine generation. The combined equation effectively averaged these opposing biases.

### 3.3. Advantages of the Combined Equation

The CKD-EPI 2021 combined equation uses both endogenous markers, allowing one to offset the limitations of the other. Its strengths are particularly evident when eGFR estimates are discordant or when the patient has confounding conditions affecting either marker. Several guidelines, including KDIGO and NICE, have endorsed its use in specific clinical situations requiring high precision, such as pre-surgical assessment, drug dosing, and transplant evaluation [19,26]. Our findings support this selective use of the combined equation, particularly when eGFR_cr_ and eGFR_cysC_ differ by more than 15 mL/min/1.73 m^2^—a threshold frequently cited in the literature as clinically relevant [20,27].

### 3.4. Limitations

This study has several limitations. First, it is a single-center, retrospective analysis. Second, we did not include a healthy control group, as our cohort consisted solely of patients referred for clinical laboratory testing. Third, measured GFR (mGFR) using exogenous markers was not available. While we used comprehensive clinical adjudication as a “real-world” reference standard, we acknowledge that this is less objective than inulin or iohexol clearance. Fourth, while we screened for BMI and clinical evidence of inflammation, we could not perform systematic quantitative measurements of muscle mass (e.g., DXA) or inflammatory markers (e.g., CRP) in all patients. Finally, our study population was predominantly Caucasian (Slovenian); therefore, these findings may not be generalizable to other ethnic groups where cystatin C or creatinine metabolism may differ. Potential hormonal influences, specifically estrogen status in post-menopausal women, which may affect creatinine generation, were also not controlled for. Regarding inflammatory status, while we systematically reviewed clinical diagnoses associated with inflammation (e.g., autoimmune vasculitis, acute infections, sepsis), quantitative inflammatory markers such as C-reactive protein (CRP) were not available for all patients at the time of eGFR testing. Therefore, we could not perform a linear correlation between CRP levels and cystatin C bias. However, in cases with documented inflammatory conditions (e.g., ANCA vasculitis, pneumonia), we observed a trend toward lower eGFR_cysC_ values, consistent with the known effect of inflammation on cystatin C generation.

### 3.5. Clinical Implications

Our findings have several important clinical implications. The observed reclassification of over half of the CKD-confirmed patients (9 out of 16, 56%) when using the combined equation indicates that equation choice is not a neutral decision—it directly impacts diagnosis, staging, treatment plans, and eligibility for specialist referral. In particular, the reclassification of 4 patients from G2 to G3a has significant consequences, as CKD diagnosis is typically established at G3 or lower, triggering more intensive monitoring, cardiovascular risk management, and adjustment of drug dosages. Similar findings have been reported by Inker et al. [19], who emphasized that the combined CKD-EPI equation improves risk prediction and reduces misclassification compared to creatinine or cystatin C alone.

Despite this, many laboratories and clinicians continue to rely solely on eGFR based on creatinine due to availability, habit, or cost concerns. However, as our results show, this practice may underestimate kidney dysfunction in certain populations—particularly in elderly, sarcopenic, or hospitalized patients—which can delay diagnosis and appropriate management.

Our results advocate for a tiered, context-driven approach to eGFR estimation:Use eGFR_cr_ as the standard screening tool in stable adults without known confounders.Consider eGFR_cr-cysC_ when higher precision is needed, or when eGFR_cr_ may be unreliable (e.g., extremes of age, body size, muscle mass).Use eGFR_cysC_ in situations of suspected AKI or early-stage CKD where creatinine remains within normal range.In cases of major discrepancy between equations (>15 mL/min/1.73 m^2^), eGFR_cr-cysC_ should be prioritized or validated with measured GFR if available.

### 3.6. Toward a Tiered Diagnostic Strategy for eGFR Use

Based on our findings and supported by current literature and guidelines ([20,28]) (KDIGO 2024; NICE 2021), we propose a tiered approach to eGFR estimation:

Primary screening: Use eGFR_cr_ (CKD-EPI 2021) in stable, ambulatory adult patients with no known confounding conditions (normal body composition, no inflammation, no medications affecting creatinine).

Contextual confirmation: If there is a clinical suspicion of misclassification, or the patient belongs to a higher-risk group (elderly, low muscle mass, inflammatory state, obesity, corticosteroid use), also calculate the following parameters:

eGFR_cr-cysC_ (preferred), or eGFR_cysC_ alone (especially in AKI or dynamic situations).

Discrepant results: When eGFR values between equations differ by ≥15 mL/min/1.73 m^2^, the combined equation should be prioritized for clinical decision-making. This threshold is consistent with KDIGO and recent studies [24]. Advanced decision-making: In complex patients (e.g., transplant candidates, chemotherapy dosing, borderline CKD staging), or if decisions carry high risk, consider confirmatory testing with measured GFR where available.

### 3.7. Practical Integration into Laboratory and Clinical Workflows

To facilitate the implementation of this tiered approach, clinical laboratories should consider:Automatic reflex testing of cystatin C when eGFR_cr_ is borderline or when requested by clinicians;Including interpretative comments in reports indicating when cystatin C-based or combined eGFR may be more appropriate;Education and collaboration with nephrology and internal medicine teams on the clinical interpretation of eGFR variability.

Furthermore, clinical decision support tools in electronic health records (e.g., flagging discrepancies between eGFR equations, offering guidance on which to use) could improve the practical use of these recommendations.

### 3.8. Alignment with Guidelines and Future Directions

Our results align with recent KDIGO 2024 recommendations, which emphasize the individualized selection of filtration markers based on patient context, and with the 2023 AACC/NKF laboratory guidance on improving equity in CKD care by recognizing limitations of creatinine in specific populations [5].

Looking ahead, broader availability of standardized cystatin C testing and integration of emerging biomarkers (e.g., beta-trace protein, beta-2-microglobulin) may further refine eGFR accuracy, particularly in high-risk or vulnerable patients. To facilitate the practical implementation of these concepts, we have summarized our recommended tiered strategy for equation selection in Table 3.

## 4. Materials and Methods

### 4.1. Study Design and Population

The study was approved by the National Medical Ethics Committee of the Republic of Slovenia (approval number: 0120-141/2025-2711-3). This retrospective observational study was conducted at the Clinical Institute of Clinical Chemistry and Biochemistry, University Medical Centre Ljubljana. Data were collected over a one-year period starting in January 2023. Inclusion criteria were adult patients (≥18 years), available S_cr_ and S_cysCr_ measurements from the same blood draw, available demographic data (age, sex), and no missing laboratory values required for eGFR calculation. Exclusion criteria included age <18 years, samples from internal validation or instrument testing, unavailable cystatin C or creatinine values, acute medical conditions with fatal outcomes, and known renal malignancies.

### 4.2. eGFR Equations

We calculated eGFR using the three standard CKD-EPI equations, which utilize age, sex, and serum biomarker concentrations to estimate filtration.

CKD-EPI 2021 Creatinine Equation (eGFR_cr_): Uses S_cr_, age, and sex. It includes separate constants ($\kappa$ and $\alpha$) for males and females to account for differences in the muscle mass generation of creatinine.CKD-EPI 2012 Cystatin C Equation (eGFR_cysC_): Uses S_cysC_, age, and sex. Unlike creatinine, this equation does not require race adjustments but relies on the constant generation of cystatin C by nucleated cells.CKD-EPI 2021 Combined Equation (eGFR_cr-cysC_): Incorporates both markers (S_cr_ and S_cysC_) along with age and sex to mitigate the non-GFR determinants affecting each marker individually.

### 4.3. eGFR Calculation and GFR Staging

For each patient, three equations were used to estimate GFR, normalized to body surface area (mL/min/1.73 m^2^):Creatinine-based (CKD-EPI 2021, eGFR_cr_);Cystatin C-based (CKD-EPI 2012, eGFR_cysC_);Combined creatinine + cystatin C (CKD-EPI 2021, eGFR_cr-cysC_).

Patients were categorized according to KDIGO GFR categories (G1–G5). The categories were defined as follows: G1 (>90 mL/min/1.73 m^2^, normal or high); G2 (60–89 mL/min/1.73 m^2^, mildly decreased); G3a (45–59 mL/min/1.73 m^2^, mildly to moderately decreased); G3b (30–44 mL/min/1.73 m^2^, moderately to severely decreased); G4 (15–29 mL/min/1.73 m^2^, severely decreased); and G5 (<15 mL/min/1.73 m^2^, kidney failure). Discrepancies between equations were defined as assignment to different GFR categories based on the equation used.

### 4.4. Selection of Clinically Relevant Subset

Patient selection and subgroup identification are summarized in Figure 3. From the 435-patient cohort, a subgroup of 44 patients showed inconsistent GFR staging between the three equations. This subgroup was selected for further evaluation. After reviewing clinical documentation in the electronic health record system, 16 patients with CKD were identified. Additional cases with AKI or UTI were analyzed to assess potential biomarker-specific interferences. Patients were assessed for clinical characteristics that could affect S_cr_ (e.g., muscle mass, trimethoprim use) or S_cysC_ (e.g., inflammation, steroid therapy, obesity). This information was used to determine which eGFR equation best reflected true renal function in each case. To validate the eGFR results, “Clinical Diagnosis” was established through a comprehensive chart review. A confirmed diagnosis of CKD was defined based on a combination of (1) longitudinal stability or decline of S_cr_ over > 3 months; (2) presence of structural kidney abnormalities on ultrasound; (3) persistent albuminuria; and (4) documented nephrology diagnosis in the electronic health records.

### 4.5. Laboratory Measurements

S_cr_ was measured using an enzymatic method on an Abbott Alinity c clinical chemistry analyzer (Abbott Laboratories, Chicago, IL, USA). The assay is traceable to the isotope dilution mass spectrometry (IDMS) reference method, in accordance with international calibration standards. S_cysC_ was determined using a particle-enhanced immunonephelometric assay (PENIA) on the Abbott Alinity c analyzer, traceable to the ERM-DA471/IFCC reference material. All analyses were performed in the Clinical Institute of Clinical Chemistry and Biochemistry of the University Medical Centre Ljubljana. Internal quality control procedures were conducted daily, and the assays fulfilled all analytical quality specifications according to ISO 15189 standards [29].

### 4.6. Statistical Analysis

All data were analyzed using GraphPad Prism (version 10.1.1, GraphPad Software, Boston, MA, USA). The normality of data distribution was evaluated using the Shapiro-Wilk test. Continuous variables with a normal distribution were compared using the paired t-test, while non-normally distributed variables were analyzed using the Wilcoxon signed-rank test. To account for multiple comparisons between the three equations (Cr vs. CysC, Cr vs. Combined, CysC vs. Combined), a Bonferroni correction was applied to the significance threshold. A *p*-value of < 0.05 was considered statistically significant.

## 5. Conclusions

This study highlights the clinical relevance of selecting appropriate eGFR equations based on patient context and the purpose of kidney function assessment. While the creatinine-based CKD-EPI 2021 equation remains a practical and widely used tool for routine estimation of GFR, our results show that it may overestimate renal function in certain patient groups, particularly the elderly and those with reduced muscle mass. The cystatin C-based equation often classified patients into more advanced CKD stages, potentially enabling earlier recognition of impaired renal function but also increasing the risk of overestimation in inflammatory states. The combined creatinine–cystatin C equation (CKD-EPI 2021) demonstrated the highest concordance with clinical diagnoses, effectively mitigating the limitations of either biomarker alone. In cases where eGFR estimates differ significantly, or where precision is essential for clinical decision-making (e.g., CKD staging, drug dosing, referral timing), the combined creatinine–cystatin C equation (CKD-EPI 2021) demonstrated improved concordance with clinical diagnoses in this cohort. These findings support a tiered approach where the combined equation is prioritized for discordant cases. Our findings support a personalized and context-driven approach to eGFR interpretation and advocate for the inclusion of cystatin C-based estimation in clinical practice, particularly when creatinine-based estimates are unreliable. Further prospective studies with measured GFR validation are needed to confirm these findings and to optimize eGFR use in diverse clinical settings.

## Figures and Tables

**Figure 1 ijms-27-00364-f001:**
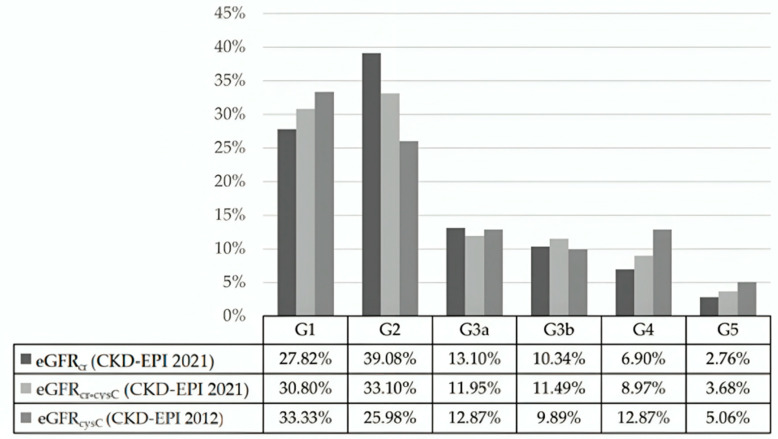
Distribution of GFR categories (G1–G5) by the eGFR equation.

**Figure 2 ijms-27-00364-f002:**
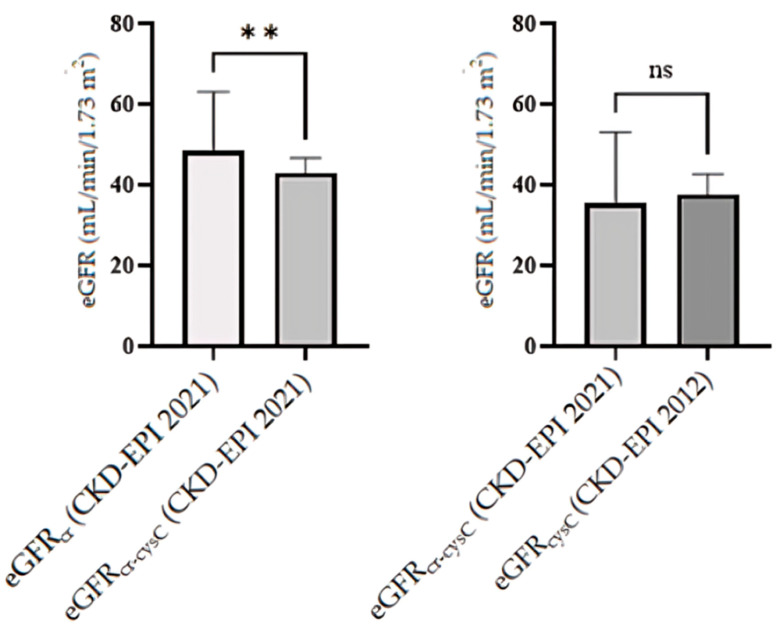
Paired comparison of eGFR values across equations in CKD patients. The data between eGFR_cr_ (CKD-EPI 2021) and eGFR_cr-cysC_ (CKD-EPI 2021) are presented using the mean and standard error of the mean (SEM) (**left**). The comparison between eGFR_cr-cysC_ (CKD-EPI 2021) and eGFR_cysC_ (CKD-EPI 2012) is shown using the median and 95% confidence interval (CI) (**right**). ** significant difference; ns non-significant difference. Bar colors indicate different eGFR equation applications.

**Figure 3 ijms-27-00364-f003:**
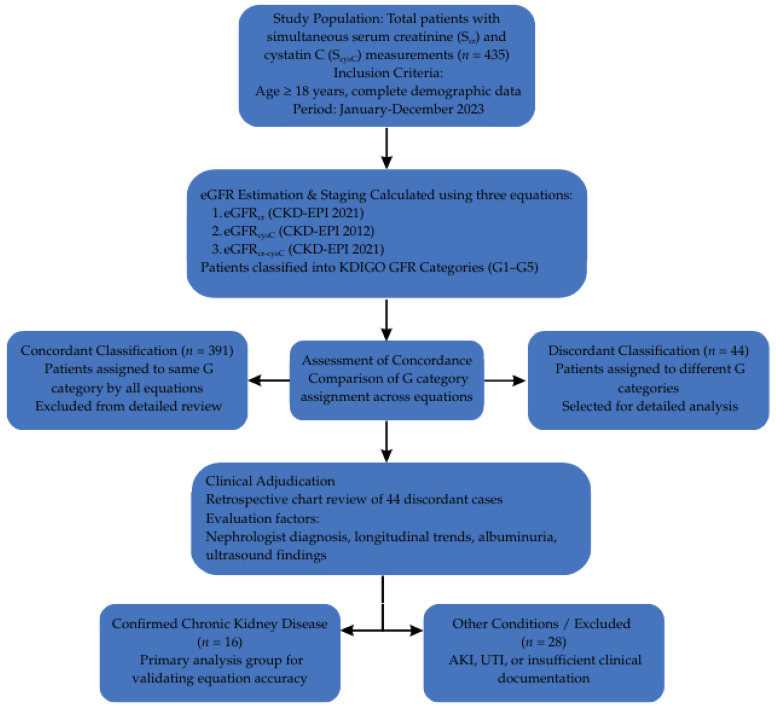
Flowchart of the study design, patient selection, and identification of the discordant subgroup. Flowchart of the study design, patient selection, and identification of the discordant subgroup. From the initial cohort of 435 patients, 44 (10.1%) showed discrepancies in GFR staging (G1–G5) between equations. These underwent detailed clinical adjudication to identify 16 confirmed CKD cases for the validation analysis.

**Table 1 ijms-27-00364-t001:** Number of individuals by glomerular filtration rate categories (G), calculated using different eGFR equations.

Equation	G1	G2	G3a	G3b	G4	G5	Total
eGFR_cr_ (CDK-EPI 2021)	121	170	57	45	30	12	435
eGFR_cr-cysC_ (CDK-EPI 2021)	134	144	52	50	39	16	435
eGFR_cysC_ (CDK-EPI 2012)	145	113	56	43	56	22	435

**Table 2 ijms-27-00364-t002:** Summary of CKD patients with GFR category discrepancies and clinical context (age, comorbidities, relevant interferences).

	Number of Patients	Sex–Female (%)	Sex–Male (%)	Age–Female (Mean)	Age–Male (Mean)	Age (Mean)
Total:	16	43.75	56.25	72	61	67
**CKD Stage Distribution:**		**CKD Stage 2 (%)**	**CKD Stage 3 (%)**	**CKD Stage 4 (%)**	**CKD Stage 5 (%)**	**CKD Unspecified (%)**
		6.25	56.25	25.00	6.25	6.25
**Equation Concordance:**		**Best Fit: eGFR_cr_ (%)**		**Best Fit: eGFR_cr-cysC_ (%)**		**Best Fit: eGFR_cysC_ (%)**
		12.50		56.25		31.25
**Clinical Characteristics:**	**Increased BMI (%)**	**Decreased BMI (%)**	**Autoimmune diseases (%)**	**Liver diseases (%)**	**Muscle mass issues (%)**	**Arterial Hypertension (%)**
	43.75	12.50	43.75	31.25	50.00	68.75
**Comorbidities:**	**Smoking (%)**		**Diabetes mellitus (%)**	**Proteinuria (%)**	**Kidney transplant (%)**	**Hypothyroidism (%)**
	18.75		37.50	56.25	18.75	18.75

BMI, body mass index; CKD, chronic kidney disease; eGFR, estimated glomerular filtration rate.

**Table 3 ijms-27-00364-t003:** Proposed tiered approach for eGFR equation selection aligned with KDIGO/NICE recommendations.

Clinical Scenario	Recommended Equation	Rationale
Routine Screening	eGFR_cr_ (CKD-EPI 2021)	Cost-effective; sufficient for stable patients with normal body composition.
Suspected Error/High Risk (Extremes of muscle mass, elderly, amputation)	eGFR_cr-cysC_	Mitigates muscle mass bias; improves accuracy.
Dynamic/Acute Changes (AKI, unstable creatinine)	eGFR_cysC_	Shorter half-life; less dependent on muscle metabolism.
Discordant Results (Diff > 15 mL/min/1.73 m^2^)	eGFR_cr-cysC_ or mGFR	Combined equation balances errors; mGFR provides a definitive answer.

## Data Availability

The data that support the findings of this study are available from the study’s principal investigator—J.O.—upon reasonable request.

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
