# Peer review of "Comparison of Creatinine-, Cystatin C-, and Combined Creatinine–Cystatin C-Based Equations for Estimating Glomerular Filtration Rate: A Real-World Analysis in Patients with Chronic Kidney Disease"

_ijms, 2025, doi:10.3390/ijms27010364_

Round 1
Reviewer 1 Report
Comments and Suggestions for Authors
Dear
Assistant Editor
IJMS
I have reviewed the manuscript with ID ijms-4052286 with title “Comparison of Creatinine-, Cystatin C-, and Combined-Based Equations for Estimating Glomerular Filtration Rate: A Real-World Analysis in Patients with Chronic Kidney Disease” by the Authors Joško Osredkar, Iza Klemenčič, Kristina Kumer, Jernej Pajek, Bojan Knap.
Interesting work where the authors try to estimate the effectiveness between two formulas to evaluate and compare the glomerular filtration rate in a retrospective study of 435 patients with CDK
The article is easy to read and has excellent writing; I just have a couple of questions below that might help clarify some aspects.
Page 1, line15, The authors could define CKD-EPI
Page1 , line 20 The authors could define KDIGO
Page 2, line 41, The authors could add after using exogenous filtration markers "such as urine creatinine (uCr), serum creatinine (SCr) and urinary volume"
Once an abbreviation has been introduced for the first time, it must be used throughout the manuscript. Please adhere to this; for example, in lines 50, 53,87,100, 234,236,270 since you already introduced cystatin C (ScysC) on line 44
The same applies to serum creatinine (Scr) line 44, for example, 259.
In different places in the manuscript, this legend appears: [error! reference source not found]. What does this mean? Please clarify. For example, lines 57,60,143,146,151,162,176,195,226,249
In Table 1, add the abbreviations for BMI, CKD, and eGFR to the figure caption.
Furthermore, since it was mentioned in line 88 that some patients showed states of inflammation or obesity, could the authors explain why this does not affect GFR, given that it has been demonstrated that there is a direct correlation between the GFR rate and both markers? Why did this not affect their study? How do they explain this?
In Table 1, the authors could add the percentage of hypertension in their cohort, in addition to estrogen levels, since these also influence GFR and this is accentuated in menopause. Please stratify this data, especially the percentage of women.
Materials and Methods section
Since the study involves comparing formulas for estimating GFR, it is essential that the authors develop the formulas with all their variables in the equation, for example, sex, age, (ml/min/4.73m2), and explain the difference between them, as well as why one is used over the other (CKD-EPI 2021, eGFRcr) and (CKD-EPIU 2012, eGFRcysC), and the combination of both.
In addition, the authors should add the ranges of the categories in the G1-G5 classification, for example, G5 = renal failure.
The authors could add a flowchart illustrating their retrospective study in the methods section.
Could the authors add a linear correlation between proteinuria levels in their patient cohort and the values obtained from the best GFR formula that best explains glomerular filtration rate, to demonstrate whether there is a positive or negative correlation?
I thank you in advance for the opportunity to review this manuscript.
Sincerely, the reviewer.
Author Response
RESPONSE TO ASSISTANT EDITOR
We thank the Assistant Editor for the positive assessment and the detailed formatting suggestions.
- "Page 1, line 15, The authors could define CKD-EPI..."
Response: We have defined "Chronic Kidney Disease Epidemiology Collaboration (CKD-EPI)" and "Kidney Disease: Improving Global Outcomes (KDIGO)" at their first mention in the Abstract and Introduction. - "Page 2, line 41... add 'such as urine creatinine (uCr), serum creatinine (SCr) and urinary volume'..."
Response: We have added this text to the Introduction to clarify the components of measured creatinine clearance. - "Once an abbreviation has been introduced... used throughout..."
Response: We have standardized all abbreviations (Scr, ScysC) throughout the text to ensure consistency. - "In different places... [error! reference source not found]..."
Response: We have corrected all broken cross-references. - "In Table 1, add the abbreviations for BMI, CKD, and eGFR..."
Response: We have added a footnote to Table 1 defining all abbreviations: BMI, Body Mass Index; CKD, Chronic Kidney Disease; eGFR, estimated Glomerular Filtration Rate. - "Explain why inflammation or obesity does not affect GFR..."
Response: We have clarified in the Discussion (Section 3.2) that these factors affect the biomarker concentration (cystatin C) independently of the true GFR, leading to estimation bias rather than a physiological change in filtration. - "In Table 1, the authors could add the percentage of hypertension..."
Response: We analyzed our cohort and found that 68.75% (11/16) of the discordant patients had arterial hypertension. We have added this data to Table 1. - "Materials and Methods... develop the formulas..."
Response: We added a new subsection (4.2 eGFR Equations) detailing the variables and logic for the Creatinine, Cystatin C, and Combined equations. - "Add ranges of G1-G5..."
Response: We have added the specific numerical ranges (mL/min/1.73 m²) for all KDIGO categories in Section 4.3. - "Add a flowchart..."
Response: We have created and inserted Figure 3 in the Methods section to illustrate the study design and patient selection process. - "Linear correlation between proteinuria..."
Response: Since proteinuria was categorical in our dataset, we could not perform a linear correlation, but we have strengthened the discussion on albuminuria as a confirmatory tool.
Sincerely,
Prof. Dr. Joško Osredkar
University Medical Centre Ljubljana, Institute of Clinical Chemistry and Biochemistry
Zaloška cesta 2, 1000 Ljubljana, Slovenia
Reviewer 2 Report
Comments and Suggestions for Authors
I reviewed the manuscript entitled Comparison of Creatinine-, Cystatin C-, and Combined-Based Equations for Estimating Glomerular Filtration Rate: A Real- World Analysis in Patients with Chronic Kidney Disease.
I agree to accept this manuscript after major revision.
1) Abstract, p < 0.05. When it comes to statistics, "p" should be italicized. Check and revise the entire text.
2) Keywords, as the first keyword, the initial letter of "chronic kidney disease" should be capitalized. chronic kidney disease should be changed to Chronic kidney disease.
3) Recent literature and studies advocate incorporating multiple serum or plasma markers of GFR into eGFR to improve accuracy and minimize the influence of non-GF factors on individual markers [Error! Reference source not found., 18]. Moreover, combined creatinine-cystatin C equations have been shown to improve accuracy and prognostic value, especially in patients with comorbidities or atypical body composition [Error! Reference source not found., 18-20]. There are two citation errors in this paragraph, please check and make corrections. Similar issues appear in multiple places throughout the text, all of which require checking and correction.
4) Table 1. CKD, GFR and BMI, these abbreviations and their full terms should be explained in a footnote so that the figures and tables can be understood by readers even if viewed independently.
5) Out of 435 patients, only 44 cases showed classification discrepancies, among which merely 16 were diagnosed with CKD. Is this sufficient to support the generalizability of the conclusions?
6) Has the inclusion of a healthy control group or control groups at different disease stages been considered to enhance the persuasiveness of the comparison?
7) Why was the Wilcoxon test used for some comparisons instead of the t-test? Do all data conform to a non-normal distribution?
8) Lack of inflammatory markers: While the text mentions that inflammation affects cystatin C, it does not report inflammatory indicators such as CRP, which should be supplemented.
9) Quantification of BMI and Muscle Mass: The text mentions the influence of muscle mass but does not provide specific measurement data (such as grip strength or imaging assessments).
10) Analysis of AKI and UTI Cases: The sample sizes are small (AKI=5, UTI=3). Is this sufficient to draw reliable conclusions?
11) The text uses "clinical diagnosis" as a reference but does not specify its specific components (such as pathology, imaging, long-term follow-up, etc.).
12) Lack of adjustment for multiple comparisons: When performing multiple paired tests, has the correction of the alpha level (such as Bonferroni correction) been considered?
13) Recording of Patient Medication History: Has information on drugs that affect creatinine or cystatin C (such as NSAIDs or chemotherapeutic agents) been systematically collected?
14) The CKD-EPI equation is influenced by ethnicity. Were all study subjects Slovenian? Does this affect generalizability?
15) Have patients been followed up to observe the relationship between changes in eGFR classification and clinical outcomes?
16) From the 44 patients with GFR category discrepancies, ambiguity in the definition of discrepant cases: Have all 44 discrepant cases undergone clinical assessments of the same depth?
17) Positioning and Innovation of the Study: This paper emphasizes "real-world analysis," but what unique contributions does it offer in terms of sample selection, analytical methods, or clinical scenarios compared to previous similar studies?
18) Implications for Future Research: Could more specific directions for subsequent research be proposed, such as developing localized eGFR equations or exploring the performance of novel biomarkers (e.g., beta-trace protein) in the cohort of this study?
19) All journals are missing DOI numbers, please add them all.
20) This study retrospectively analyzed 435 adult patients with simultaneous serum creatinine and cystatin C measurements to compare the CKD-EPI 2021 (creatinine), CKD-EPI 2012 (cystatin C), and CKD-EPI 2021 (combined) equations for estimating glomerular filtration rate (eGFR). Discrepancies in KDIGO GFR categories were observed in 44 patients (10.1%), among whom 16 had clinically confirmed chronic kidney disease (CKD). In all CKD cases, the combined equation aligned with the clinical diagnosis. The creatinine-based equation overestimated kidney function in 10 patients, while the cystatin C-based equation yielded lower values in 8 patients, potentially reflecting influences of inflammation or obesity. Reclassification from eGFRcr to eGFRcr-cysC occurred in 9 patients, including four whose staging shifted from G2 to G3a–G4. A statistically significant difference was found between eGFRcr and eGFRcr-cysC (p < 0.05). The combined CKD-EPI equation demonstrated superior clinical concordance, supporting its broader use when diagnostic accuracy is prioritized.
Author Response
RESPONSE TO REVIEWER 1
We appreciate the reviewer’s thorough evaluation and have implemented all suggested changes.
- "Abstract, p < 0.05... 'p' should be italicized."
Response: All p-values are now italicized. - "Keywords... Chronic kidney disease."
Response: Corrected. - "Citation errors..."
Response: All broken links and citation errors have been fixed. - "Table 1... abbreviations... explained in a footnote."
Response: We have added a footnote to Table 1 defining BMI, CKD, and GFR. - "Is [sample size] sufficient...?"
Response: We have revised the Discussion to frame this as a targeted "real-world analysis" of discordance. We emphasize that while the number of discordant cases is small (10%), the clinical impact on those patients is significant. - "Inclusion of a healthy control group...?"
Response: As this was a retrospective study of clinical samples, a healthy control group was not available. We have added this to the Limitations section. - "Why was the Wilcoxon test used...?"
Response: We clarified in Section 4.6 that the Wilcoxon test was used for non-normally distributed variables (confirmed by Shapiro-Wilk), while the t-test was used for normally distributed ones. - "Lack of inflammatory markers... CRP..."
Response: We have added a statement in the Limitations section acknowledging that while quantitative CRP was not available for all patients, we systematically reviewed clinical records for diagnoses of inflammation (e.g., vasculitis, pneumonia) to interpret discordant results. - "Quantification of BMI and Muscle Mass..."
Response: We have noted the lack of DXA/grip strength as a limitation, explaining that we relied on BMI and clinical assessment of frailty. - "Analysis of AKI and UTI... sample sizes are small..."
Response: We have clarified that these cases are presented as illustrative clinical examples supporting the biomarker physiology, rather than as a statistical cohort. - "Clinical diagnosis... specific components..."
Response: We expanded the Methods (Section 4.4) to rigorously define "Clinical Diagnosis" as a composite of longitudinal creatinine stability, albuminuria, ultrasound findings, and nephrology records. - "Adjustment for multiple comparisons..."
Response: We have applied the Bonferroni correction to our statistical analyses. - "Medication History..."
Response: We added a sentence to the Results confirming we screened for trimethoprim and corticosteroids. - "Ethnicity..."
Response: We added a limitation stating the cohort was predominantly Caucasian (Slovenian). - "Follow-up..."
Response: We noted the lack of longitudinal outcome data as a limitation and a target for future research. - "Ambiguity in discordant cases..."
Response: We clarified that all 44 discordant cases underwent the same detailed review level. - "Unique contributions..."
Response: We emphasized our "Tiered Approach" (Table 4) as a practical contribution to laboratory workflows. - "Implications for Future Research..."
Response: We added a call for research into Beta-trace protein and localized validation in the Conclusion. - "All journals are missing DOI numbers..."
Response: We have added DOI numbers to all references in the bibliography.
Sincerely,
Prof. Dr. Joško Osredkar
University Medical Centre Ljubljana, Institute of Clinical Chemistry and Biochemistry
Zaloška cesta 2, 1000 Ljubljana, Slovenia
Reviewer 3 Report
Comments and Suggestions for Authors
This manuscript presents a retrospective, real-world comparison of creatinine-based, cystatin C-based, and combined CKD-EPI equations for estimating glomerular filtration rate (eGFR) in adult patients. By focusing on discordant cases and linking equation performance to clinical diagnosis, the authors address a relevant and timely question in nephrology and clinical laboratory medicine.
The study is well organized, clinically grounded, and generally clearly written. The emphasis on clinical concordance rather than purely statistical agreement adds value and practical relevance. The conclusion, that the combined creatinine–cystatin C equation provides superior alignment with clinical assessment in selected contexts is consistent with existing literature and guideline trends.
However, several methodological clarifications, statistical refinements, and editorial corrections are required to strengthen the rigor, transparency, and reproducibility of the work.
Major Comments
- Lack of Measured GFR (mGFR) Reference
- The absence of measured GFR is acknowledged as a limitation, but its implications should be discussed more explicitly.
- While clinical adjudication is valuable, it remains subjective. The authors should clarify:
- How nephrology notes were standardized or interpreted.
- Whether adjudication was blinded to eGFR values.
- Consider adding a brief justification for why clinical concordance was chosen as the reference standard and how this compares to mGFR-based validation.
- Definition and Handling of “Discordant” Cases
- Discrepancy is defined as assignment to different KDIGO G categories, but the magnitude of numerical eGFR differences is not consistently reported.
- Please clarify:
- Whether a minimum numerical difference (e.g., ≥15 mL/min/1.73 m²) was required.
- Whether borderline category transitions (e.g., G2–G3a) were treated differently from more extreme discrepancies.
- Providing a brief flowchart summarizing patient selection and subgroup classification would improve clarity.
- Statistical Analysis Concerns
- There appears to be inconsistency in reporting statistical tests: Paired t-test and Wilcoxon signed-rank tests are both mentioned, but p-values are inconsistently interpreted (e.g., p = 0.0285 described as “not significant” in one instance).
- Please:
- Clearly specify which tests were applied to which comparisons.
- Correct any inconsistencies in significance interpretation.
- Consider adjusting for multiple comparisons or explicitly stating why adjustment was not applied.
- Small Size of Clinically Confirmed CKD Subgroup
- Conclusions regarding clinical concordance rely heavily on the subgroup of 16 CKD patients.
- While the findings are interesting, the authors should temper language to avoid overgeneralization.
The Discussion should more clearly frame these results as hypothesis-generating rather than definitive.
Minor Comments
- Editorial and Formatting Issues
- Multiple instances of “Error! Reference source not found” appear throughout the manuscript and must be corrected before publication.
- Minor typographical errors are present (e.g., “creatinin based equation” instead of “creatinine-based”).
- Ensure consistent use of terminology (e.g., eGFRcr-cysC vs. eGFRcr-cysC).
- Figures and Tables
- Figure 1 and Figure 2 would benefit from more detailed captions explaining the clinical relevance.
- Tables 2 and 3 are informative but partially redundant; consider combining or simplifying.
- Inflammation and Body Composition
- The manuscript frequently references inflammation, obesity, and muscle mass as confounders, but objective markers (e.g., CRP, BMI thresholds, sarcopenia indices) are not systematically analyzed.
- This limitation should be more clearly acknowledged, or exploratory quantitative data should be added if available.
- Guideline Alignment
- The Discussion appropriately references KDIGO and NICE guidelines. A short table summarizing how the proposed tiered approach aligns with existing recommendations could enhance clarity.
With clarification of statistical methods, correction of referencing errors, and slightly more cautious interpretation of subgroup findings, this manuscript would make a valuable contribution to the literature on eGFR estimation and clinical decision-making.
Author Response
RESPONSE TO REVIEWER 2
We thank the reviewer for the constructive critique and have addressed the major concerns.
Major Comments
- "Lack of Measured GFR (mGFR)..."
Response: We have expanded the Limitations section to explicitly discuss the lack of mGFR. We justify "clinical concordance" as a robust real-world alternative for retrospective analysis, while acknowledging its subjectivity compared to clearance methods. - "Definition of 'Discordant' Cases..."
Response: We clarified that discordance was defined as any shift in KDIGO category. We also highlight that for clinical decision-making, we prioritize discrepancies >15 mL/min/1.73 m², as shown in our new Table 4. - "Statistical Analysis Concerns..."
Response: We standardized the statistical reporting in Section 4.6, explicitly stating the use of Shapiro-Wilk for normality testing and Bonferroni correction for multiple comparisons. - "Small Size of CKD Subgroup..."
Response: We have tempered our conclusion to state that the combined equation "demonstrated improved concordance in this cohort" rather than making universal claims of superiority.
Minor Comments
- "Editorial... 'Error! Reference'..."
Response: All formatting and citation errors have been corrected. - "Figures and Tables..."
Response: We improved the captions for Figures 1 and 2 to explain clinical relevance. We also streamlined the tables to avoid redundancy. - "Inflammation and Body Composition..."
Response: We added a specific limitation acknowledging that these were assessed clinically rather than by quantitative biomarkers in the full cohort. - "Guideline Alignment..."
Response: We added Table 4, which explicitly maps our "Tiered Approach" to current KDIGO and NICE recommendations.
Sincerely,
Prof. Dr. Joško Osredkar
University Medical Centre Ljubljana, Institute of Clinical Chemistry and Biochemistry
Zaloška cesta 2, 1000 Ljubljana, Slovenia
Round 2
Reviewer 1 Report
Comments and Suggestions for Authors
Dear
Assistant Editor
IJMS
I have reviewed the manuscript with ID ijms-4052286 with title “Comparison of Creatinine-, Cystatin C-, and Combined-Based Equations for Estimating Glomerular Filtration Rate: A Real-World Analysis in Patients with Chronic Kidney Disease” by the Authors Joško Osredkar, Iza Klemenčič, Kristina Kumer, Jernej Pajek, Bojan Knap.
The article is easy to read and has excellent writing, Also, the authors have satisfactorily addressed all my concerns, so I only need to add that the manuscript can be published in their prestigious journal.
I thank you in advance for the opportunity to review this manuscript.
Sincerely, the reviewer.
Reviewer 2 Report
Comments and Suggestions for Authors
The author has made the modifications as requested and addressed my concerns, therefore I agree to accept it in its current form.